# ERROR DYNAMICS OF SYMBOLIC CONTEXT IN SMALL TRANSFORMERS

## ABSTRACT

Language models often recover from partial corruption in their inputs, yet the mechanism behind this "spontaneous context restoration" is unclear. We study controlled, label-preserving corruptions in symbolic arithmetic and find a consistent mid-to-late-layer elbow where later components integrate surviving cues to reconstruct the answer. We introduce two readouts, Repair Difference (RD), a logit-space contribution measure, and Token Agreement (TA), a layer-wise consistency score, and a linearity-scale test that predicts repairability. We find near-linear behavior on clean inputs and pronounced nonlinearity under corruption; the linearity residual predicts repair success. Across model families, accuracy degrades smoothly with corruption ($\rho \approx -1$) and yields compact robustness summaries ($\tau_{50} \approx 27\text{–}34\%$). RD/TA peak near the elbow, localizing where repair occurs. Brief fine-tuning at moderate corruption improves self-repair, whereas training on heavy corruption weakens it, giving a simple, data-efficient recipe. To test the linearity claim beyond arithmetic, we replicate the context perturbation correlation to the local non-linearity in the NLP corruption task. Together, RD, TA, $\tau_{50}$, and the linearity test form a concise toolkit for diagnosing and training for spontaneous context restoration on corrupted contexts and actionable guidance for when and how models repair corrupted context, offering practical levers for debugging, evaluation, and training.

## 1 INTRODUCTION

Small transformers are increasingly used in settings where inputs are imperfect, such as on-device inference, latency-sensitive assistants, and symbolic subroutines where a single mistyped digit can derail a computation (Sanh et al., 2019; Sun et al., 2020; Magister et al., 2022; Wang et al., 2024). Despite anecdotal evidence that models sometimes "snap back" from such noise, we lack a concrete account of *when* **spontaneous context restoration** happens and *where* in the network it is implemented, connecting to recent accounts of symbol processing and variable binding in in-context learning (Smolensky et al., 2024). Prior interpretability has mapped arithmetic circuits under clean inputs (Elhage et al., 2021; Nanda et al., 2023), while robustness studies often perturb the surrounding text rather than the symbolic content that defines the task. We address this gap by directly corrupting the numbers in controlled arithmetic sequences and tracing the computations that support recovery. Aligning with Cheng et al. (2022) on symbolic context, but differing in that we analyze in-network repair dynamics without constraining outputs to a formal language.

Our study introduces three label-preserving corruption regimes (zero, in-range, out-of-range) and two readouts that expose repair dynamics: *Repair Difference* (RD), a logit-space measure of how strongly the model pushes the correct token on corrupted inputs, and *Token Agreement* (TA), a layer-wise consistency score with the clean target. Together they reveal a consistent *mid-to-late elbow*: early layers largely transport corrupted evidence, while later layers integrate the surviving cues to reconstruct the intended pattern. Practically, this depth profile translates this spontaneous context restoration from an observed behavior into an actionable diagnostic one.

We further conducted a linearity–scale test that perturbs hidden states from clean toward corrupted along the observed error direction and compares first-order predictions to actual logit changes. The root mean squared error, which demonstrates how nonlinear the mapping is locally, predicts whether the correction will succeed. This connects a mechanistic signal (local curvature) to an operational

one (accuracy under corruption): when behavior is locally linear, models are predictable and repairable; when curvature grows (typically in late layers and under stronger corruption), errors amplify. We replicated this behavior on a natural language task and showed that local linearity extrapolates to Large Language Models during this correction. Finally, a brief fine-tune on moderate corruption improves robustness broadly, whereas training on heavy corruption over-specializes and weakens context correction. This suggests a complementary path to robustness alongside parameter-efficient masking approaches Zhao et al. (2020). For practitioners, this yields a compact toolkit (RD, TA, $\tau_{50}$, linearity residual) and a simple recipe: probe for a mid/late repair step, keep behavior in the near-linear regime, and calibrate noise during fine-tuning to strengthen.

## 2 RELATED WORK

### 2.1 INTERPRETING ARITHMETIC AND MATHEMATICAL OBJECTIVES

Prior work on arithmetic, algebra, and symbolic prediction (Yu & Ananiadou, 2024; Zhang et al., 2024; Stolfo et al., 2023; Peng et al., 2022; Brinkmann et al., 2024) shows that a small set of attention heads and residual paths encode rules, while MLPs compose intermediate features; these circuits can be localized, traced, and even edited. However, most analyses assume clean inputs, whereas real prompts often contain omissions, spurious tokens, or adversarial substitutions. We therefore ask: (i) how do internal circuits respond to exogenous corruption, and (ii) can models dynamically realign to recover the intended pattern? Using controlled input ablations and mechanistic probes on small transformers, we test how context restoration unfolds when context is pathologically perturbed. Finally, we show that fine-tuning with moderate corruption produces the strongest robustness gains, clarifying how training noise should be calibrated for reliable symbolic reasoning. In comparison, PoT and NeRd improve numerical reasoning by delegating computation to an external executor or program, our results characterize where such computation-like effects arise internally (Chen et al., 2022; 2019).

### 2.2 PROMPT PERTURBATION

Prior work tests reasoning reliability by perturbing the prompt, chiefly with textual/contextual noise—e.g., extra punctuation, irrelevant/adversarial descriptions, or extraneous numbers—finding smooth degradation with noise and partial mitigation via prompting or fine-tuning (e.g., contrastive denoising CoT) (Abedin et al., 2025; Chatziveroglou et al., 2025; Anantheswaran et al., 2024; Zhou et al., 2024). Yet these studies rarely corrupt the numeric content itself, altering the wrapper rather than the numbers that define structure. We instead directly corrupt the numerical context in symbolic sequences (zero ablation; in-range and out-of-range substitutions) and link outcome changes to internal repair dynamics, yielding mechanistic accounts of spontaneous context correction rather than aggregate robustness alone.

### 2.3 MECHANISTIC METHODS

General-purpose mechanistic tools—activation patching and residual probing—map causal pathways in transformers (Elhage et al., 2021; Nanda et al., 2023). Circuit discovery and sparse autoencoders recover feature-level structure behind complex behaviors (Olah et al., 2020; Bricken et al., 2023; Heap et al., 2025), forming the basis for our layer-wise probes and repair metrics. Beyond math, LLMs are brittle to adversarial prompts, formatting shifts, and subtle context changes (Zou et al., 2023; Li et al., 2023). The Hydra Effect shows endogenous self-repair via redundant pathways that preserve function under ablations (McGrath et al., 2023). We study interpretability jointly: directly corrupting numeric content (not just surrounding text) and analyzing how small transformers internally realign to recover symbolic structure.

## 3 METHOD

### 3.1 TASK

We leverage counting tasks with metered steps to evaluate error dynamics. We instantiated three families- *constant*, *subtract*, and *variable-step*.

**Sequence families.** For each family, let $p$ be the starting number, $q$ the (base) step size, $r$ the last shown element, and $L$ the number of displayed terms (so $r = s_L$). We draw real-valued samples $p^* \sim \mathcal{N}(\mu_p, \sigma_p^2)$ and $q^* \sim \mathcal{N}(\mu_q, \sigma_q^2)$, then set the integer parameters

$$p = \lfloor |p^*| \rfloor, \qquad q = \max\{1, \, \lfloor |q^*| \rfloor\}.$$

Indices run over $i \in \{1, \dots, L\}$, and values obey $0 \leq s_i \leq S_{\max}$, where $S_{\max}$ is a global cap.

1. **Constant (fixed increment)**:

$$S_L^{\text{const}} = \big(p, \, p+q, \, p+2q, \, \dots, \, p+(L-1)q\big), \quad r = p + (L-1)q.$$

The target next element is $r + q$.

2. **Variable-arithmetic (variable increment)**:

$$s_0 = p, \qquad s_{t+1} = s_t + (q+t) \text{ for } t = 0, \dots, L-2,$$

yielding

$$S_L^{\text{var}} = (s_0, \dots, s_{L-1}), \quad r = s_{L-1}.$$

The next increment after $r$ is $q + (L-1)$, so the target is $r + \big(q + (L-1)\big)$.

3. **Subtract (fixed decrement)**:

$$S_L^{\text{sub}} = \big(p, \, p-q, \, p-2q, \, \dots, \, p-(L-1)q\big), \quad r = p - (L-1)q,$$

with $q \geq 1$ and all terms constrained to $R$. The target is $r - q$.

**Prompt construction.** Prompts are rendered as a comma-separated list of the $L$ shown integers. We used variable lengths (excluding commas)

$$L \in \{40, \, 60, \, 80, \, 100, \, 120, \, 140\}.$$

**Ablation scheme.** To probe robustness, we randomly mask a fraction of the displayed numbers. For an ablation rate $\alpha \in \{10, 20, 30, 40, 50, 60, 70, 80, 90\}\%$, we sample without replacement a set of indices $M \subset \{1, \dots, L-1\}$ with $|M| = \lceil \alpha L/100 \rceil$, and ablate the entries at positions in $M$. The last shown element (the $L^{\text{th}}$ term, $r$) is *never* ablated. We consider three regimes: (i) *zero ablation*, replacing selected entries with the literal token 0; (ii) *in-range ablation*, replacing with integers sampled from $(\min S, \max S)$ but not equal to the ground-truth value; and (iii) *out-of-range ablation*, replacing with integers drawn from $[0, \min S) \cup (\max S, S_{\max}]$.

## 3.2 MODEL TRAINING AND PRE-TRAINED MODELS

To study error dynamics on arithmetic series, we first evaluated three off-the-shelf decoder-only LMs: *DistilGPT2* (Radford et al., 2019; Sanh et al., 2019), *Pythia-14M*, and *Pythia-70M* (Biderman et al., 2023), under a next-token prediction objective with teacher forcing (Williams & Zipser, 1989). These baselines use mixed text–number byte-level BPE tokenizers learned on broad web corpora rather than targeted arithmetic supervision. Concretely, *DistilGPT2* uses the 50,257-item GPT-2 tokenizer, while the Pythia models use the GPT-NeoX tokenizer trained on The Pile; in all cases, many multi-digit integers decompose into multiple subword units.Because such tokenizers segment integers irregularly (there is no fixed largest single-token integer and nearby values may split differently), we constrain targets so that every displayed integer is a *single* token under all three tokenizers. Empirically, the smallest safe cap across models fell between 361 and 567; we therefore set $S_{\max} = 360$ to preclude multi-token spillover.

Moreover, these models were either trained on OpenWebTextCorpus (*DistilGPT2*) Hugging Face or the Pile (*Pythia suite*), which might include noisy or inconsistently formatted numeric content; error dynamics cannot be claimed with certainty. Therefore, we trained a *decoder-only, attention-only* 4-layer transformer with $\sim$1.74M parameters on clean sequences drawn from the three arithmetic families, using the identical next-token objective. Given an input sequence $S = (s_1, \dots, s_L)$, the model is trained to maximize $p_\theta(s_{t+1} \mid s_{\leq t})$ and we report losses on the standard one-step shift. For the pre-trained setup, we set $S_{\max} = 13{,}000$.

The training corpus uses fixed sequence length $T{=}150$ and a vocabulary of 13,007 tokens consisting of the integer symbols, a comma token, and standard specials (e.g., `[PAD]`, `[EOS]`). We train on 38,867 examples. To avoid leaking numeric structure via token IDs, we randomly permuted the integer token indices at tokenizer construction time; consequently, the model must learn each numeral's value from context rather than from an ordered embedding index. Unless otherwise noted, all results for the custom model are obtained by training it independently with 5 different random seeds. The resulting accuracy curves and diagnostic plots are visually indistinguishable across seeds, so we report a representative run in the main figures.

### 3.3 DEFINING SPONTANEOUS CONTEXT CORRECTION

In the current study, we re-conceptualized the effect due to context restoration, as compared to previous studies where logit difference has been extensively used Wang et al. (2022); McGrath et al. (2023), to capture the most significant change in model behaviour. For a token sequence $S_n$, each layer's output $y_\ell$ is added to the residual stream $R_\ell$ (with $\ell$ the layer index and $n$ the prompt length), so that $R_\ell = \sum_{i=0}^{l} y_i$. In a transformer, applying LayerNorm to the residual and then the unembedding matrix $W_u$ yields the logits; at the final layer $L$ this gives $\text{Logit}_f = \text{LayerNorm}(W_u \cdot R_L)$. Analogously, we can read out intermediate predictions either from the residual, $\text{Logit}_\ell^{\text{resid}} = \text{LayerNorm}(W_u \cdot R_\ell)$, or from a single layer's contribution, $\text{Logit}_l^{\text{output}} = \text{LayerNorm}(W_u \cdot y_\ell)$. Following common practice, we mean-center logits (subtract their average) so they have zero mean, which leaves probabilities unchanged Elhage et al. (2021); Rushing & Nanda (2024); McGrath et al. (2023).

**Definition 1:** *(Logit Difference A and B)* To compute the difference in internal dynamics during the inference of clean prompts and faulty prompts, logit distributions are computed. Let $Logit$ be the logit distribution and $clean_t$ be the next maximum likelihood token during the clean prompt inference and $Logit'$ logit distribution and $ablate_t$ be the next maximum likelihood token during faulty prompts. Then logit difference is defined as the difference between $Logit'_{clean_t}$ and $Logit_{clean_t}$, $LogitDifference = Logit'_{clean_t} - Logit_{clean_t}$. However, to be able to produce the same token as that of the clean task or when self-correction takes place, $Logit'clean_t$ should be equal to $Logit'_{ablated_t}$. We termed the difference between the latter logit difference B ($LD_B$) and the former as logit difference A ($LD_A$).

$$LD_A = \text{Logit}'_{\text{clean}_t} - \text{Logit}_{\text{clean}_t}$$
$$LD_B = \text{Logit}'_{\text{ablated}_t} - \text{Logit}'_{\text{clean}_t}$$

**Definition 2**: *(Repair Difference)* The repair difference ($RD$) is defined to assess how effectively a model can compensate for changes or disruptions, such as those introduced by a faulty prompt. It is used to quantify the model's ability to self-repair or recover from errors. Since $LD_A$ measures the difference between the logits of the clean prompt and the faulty prompt after applying any corrections or adjustments and $LD_B$ measures the difference between the logits of the faulty prompt before and after applying corrections, $RD$ should be able to identify the greatest extent of correction needed to match or surpass the original clean performance. Therefore we define repair difference as the maximum distance that $Logit'_{clean_t}$ needs to cover to reach the same value as $max(Logit_{clean_t}, Logit'_{ablated_t})$.

Then repair difference $RD$ is given as,

$$RD = \begin{cases} LD_A & \text{for } Logit_{clean_t} \geq Logit'_{ablated_t} (i) \\ LD_B & \text{for } Logit'_{ablated_t} > Logit_{clean_t} (ii) \\ LD_A & \text{for } Logit'_{ablated_t} = Logit'_{clean_t} > Logit_{clean_t} (iii) \end{cases}$$

where the third case denotes when over-correction takes place.

The range of $LD_A$ is $R : (-\infty, +\infty)$, $LD_B$ is $R : [0, +\infty)$ and for repair difference $R : (-\infty, +\infty)$. $RD > 0$ iff $Logit'_{ablated_t} = Logit'_{clean_t}$ and $Logit'_{clean_t} > Logit_{clean_t}$ which indi-

cates that the model on faulty prompts is outperforming the one with clean inputs. Furthermore, like $LD_A$ and $LD_B$, $RD$ can be computed for each layer and its evolution with time from the residual.

**Definition 3 (Layer-wise token agreement).** For each layer $\ell$ and readout $\tau \in \{\mathrm{resid}, \mathrm{output}\}$, let $\mathrm{Logit}_\ell^{(\tau)\prime}$ denote the (mean-centered) logits read from layer $\ell$ on the *faulty* run, and let $\mathrm{clean}_t$ be the clean-run target token (Def. 1). The agreement is the fraction of examples for which the faulty run already predicts the clean target:

$$\mathrm{TA}_\ell^{(\tau)} \;=\; \frac{1}{N} \sum_{i=1}^{N} \mathbf{1}\left[ \mathrm{clean}_{t,i} = \arg\max \mathrm{Logit}_{\ell,i}^{(\tau)\prime} \right].$$

### 3.4 Linearity–scale test

To test the local linearity of the mapping from a layer's hidden state to the target next-token logit, we run the model on two versions of the same example: a clean input and a corrupted input. At all layers of interest, we cache the hidden states for both runs.

Let a sequence of length $L$ be processed by a model. At layer $\ell$, denote the hidden states on a *clean* input by $h_\ell^{\mathrm{clean}} \in \mathbb{R}^{L \times C}$, and on a *corrupted* input by $h_\ell^{\mathrm{corr}} \in \mathbb{R}^{L \times C}$.

Fix the predicted next-token position $p := L+1$. Let the *target* logit refer to a designated next token (e.g., the ground-truth label or the clean-run argmax). Define the *corruption direction*

$$e \;=\; h_\ell^{\mathrm{corr}} - h_\ell^{\mathrm{clean}}, \qquad e_p \in \mathbb{R}^C.$$

Let $g$ map layer-$\ell$ hidden states to the target logit; thus $g(h_\ell)$ is the scalar logit of the target token when the forward computation proceeds from the state $h_\ell$. On the clean pass, take the gradient at position $p$ $\nabla g\big(h_\ell^{\mathrm{clean}}\big)_p \in \mathbb{R}^C$.

For $\alpha \in \{0.25, 0.5, 0.75, 1.0\}$, consider the perturbed (patched) state $\tilde{h}_\ell(\alpha) = h_\ell^{\mathrm{clean}} + \alpha e$, which changes only position $p$ by $\alpha e_p$ and leaves all other positions unchanged. By the first-order Taylor expansion of $g$ around $h_\ell^{\mathrm{clean}}$,

$$g\big(\tilde{h}_\ell(\alpha)\big) \;\approx\; g\big(h_\ell^{\mathrm{clean}}\big) \;+\; \alpha\,\nabla g\big(h_\ell^{\mathrm{clean}}\big)_p^\top e_p.$$

Therefore, the predicted logit change is

$$\Delta g_{\mathrm{pred}}(\alpha) \;=\; g\big(\tilde{h}_\ell(\alpha)\big) - g\big(h_\ell^{\mathrm{clean}}\big) \;\approx\; \alpha\,\nabla g\big(h_\ell^{\mathrm{clean}}\big)_p^\top e_p.$$

The true change is

$$\Delta g_{\mathrm{true}}(\alpha) \;=\; g\big(\tilde{h}_\ell(\alpha)\big) - g\big(h_\ell^{\mathrm{clean}}\big).$$

Taylor's theorem about $h_\ell^{\mathrm{clean}}$ yields

$$\Delta g_{\mathrm{true}}(\alpha) \;=\; \alpha\,\nabla g^\top e \;+\; \tfrac{1}{2}\alpha^2\, e^\top H e \;+\; \mathcal{O}(\alpha^3),$$

where $H$ is the Hessian of $g$ with respect to $h_\ell$ at $h_\ell^{\mathrm{clean}}$. Thus, small $\alpha$ probes linearity, while larger $\alpha$ reveals curvature along $e$. Comparing $\Delta g_{\mathrm{pred}}(\alpha)$ to $\Delta g_{\mathrm{true}}(\alpha)$ across these $\alpha$ values quantifies the local linearity of the mapping at layer $\ell$ along direction $e$.

## 4 Experiments and Results

### 4.1 Performance degrades with input corruption

We evaluated 16k sequences for each of the ablation regimes. For the custom-trained model, across all regimes and sequence types, accuracy declines smoothly and near-monotonically as ablation increases (Spearman $\rho \in [-0.988, -1.000]$, all $p \leq 9.3 \times 10^{-8}$). The average accuracy across the ablation sweep for all regimes sits in the mid-30s (%). The $\tau_{50}$ (ablation at 50% accuracy) concentrates around $\sim$27–34%. Early in the sweep (0–40% ablation), where the curve is almost

linear, the accuracy drops at roughly 1.7–2.1 points per 1% ablation. Small, consistent nuances emerge by regime. For instance, SUBTRACT is slightly more robust with zero ablation (higher mean accuracy and $\tau_{50}$), CONSTANT leads for in-range ablation, and all sequence types are effectively comparable out-of-range (overlapping CIs). In short, degradation with ablation is predictable and largely uniform, with only modest, regime-specific differences in robustness. See Figure 1.

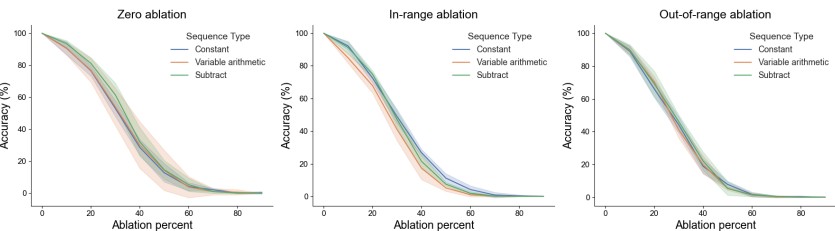

Figure 1: Accuracy of the custom model on three ablation regimes for all three sequence types.

For pre-trained models (Pythia suite and DistilGPT2), we evaluated 150 sequences each for CONSTANT and SUBTRACT sequences only, as low $S_{\max}$ constraints creating enough VARIABLE-ARITHMETIC number series. Similar to the custom model, the accuracy drops smoothly and near-monotonically as ablation increases. Overall robustness (the average accuracy across the ablation sweep) is highest for DistilGPT-2, moderate for Pythia-70M, and lowest for Pythia-14M. DistilGPT-2 maintains mid-30s to mid-40s% average accuracy with $\tau_{50}$ around 42–47%; Pythia-70M is lower (average accuracy about 27–36%, $\tau_{50}$ about 22–32%); Pythia-14M is brittle (average accuracy about 1–16%, $\tau_{50} = 0\%$). CONSTANT sequences are consistently sturdier than SUBTRACT, and smaller models show steeper early-regime slopes, reflecting faster degradation.

### 4.2 INTERNAL CORRECTION PEAKS AROUND THREE-QUARTERS DEPTH

To quantify internal correction, the model applies to overcome faulty input and still predict the right token, we computed Repair Difference (RD) per layer before and after the residual is added. RD is the shift in the correct–token logit on the ablated prompt needed to beat the harder of the two references—either the clean correct–token logit or the ablated alternate–token logit. Intuitively, RD measures how much the network "pushes back" against the corruption. For the custom-trained, RD shows a consistent elbow (See Figure 2a) at approximately three–quarters depth across tasks: a one–break segmented fit places the elbow at layer 3 of 4 (normalized depth $\approx 0.75$) for Constant, Subtract, and Variable–arithmetic sequences (See Figure 8). Before the elbow, RD changes slowly (mid–layer slopes per layer: CONSTANT $-0.65$, SUBTRACT $-0.53$, VARIABLE ARITHMETIC $-0.47$), indicating gradual damping. Immediately after the elbow, RD drops steeply (late–segment slopes large in magnitude), reflecting a decisive late–stage correction failure concentrated in the final block. In comparison, the layer-wise token agreement (TA) reveals the same depthwise pattern across tasks (See Figure 2b and Figure 9). Agreement falls sharply from layer 1 to the mid network (the strongest drop by layer 2), indicating that early processing propagates the ablation's effect. It then *recovers* toward layer 3, where agreement peaks, consistent with a mid/late *self-repair step*. The final layer contributes little additional recovery (and can slightly reduce agreement), suggesting that most correction is completed just before the output layer. Heavier ablations uniformly lower the curves.

Similarly, for the pretrained models RD exhibits the qualitative shape—a mid–to–late *elbow* followed by a sharp drop—but the location and magnitude vary by architecture. Overall depthwise ordering is preserved, indicating that pretrained LMs retain the same "mid-damping, late correction" propagation law even without task-specific training. TA is V-shaped: agreement dips sharply at a mid/late block and then rebounds to near one by the final layer. Early decline, a mid-depth failure point, and late recovery—placing the effective self-repair step just before the output layer (See Figure 8 and Figure 9).

This reveals a simple propagation law: layers at mid to three-quarters depth carry the repair signal in their residual stream, damping errors upstream and triggering a concentrated correction just before the output layer.

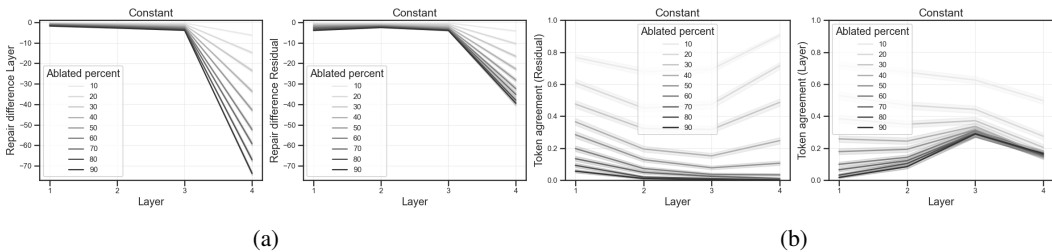

Figure 2: (a) Exemplary Repair Difference across ablation percentages for CONSTANT sequence for the custom model. (b) Exemplary Layer-wise Token Agreement (TA) across ablation percentages for CONSTANT sequence for the custom model.

We also observed some instances of *over-correction* when the repair difference is positive, i.e., RD $> 0$ (the ablated-prompt correct-token logit exceeds both the clean baseline and the ablated alternate) (See Figure 10). Over-correction is rare and concentrated in the earliest layer. Across regimes, the peak rate always occurs at layer 1 and remains modest. Heatmaps show a monotonic decline with depth. Over-correction also diminishes as ablation strength increases, indicating that mild corruptions sometimes induce small early overshoots, whereas stronger corruptions predominantly lead to under-correction. However, this trend does not hold for pre-trained models and showed higher levels, sometimes even peaking at 100%.

### 4.3 LOCAL LINEARITY PREDICTS REPAIRABILITY UNDER CORRUPTION

To assess local linearity at layer $\ell$, we compare $\Delta_{\text{true}}$ and $\Delta_{\text{pred}}$ over several $\alpha$ values. As a simple predictor at $\alpha=1$ negative Root Mean Squared Error (-RMSE), and evaluated how well $S$ separates *correct* vs. *incorrect* continuations by computing ROC-AUC within each sequence type and ablation regime. The AUC values ranged $0.91 - 0.94$ for layers except the last layer. However, within ablation levels, the AUC drops significantly to $0.4 - 0.5$, revealing Simpson's paradox. Pairwise AUC heatmaps revealed that most of the pooled AUC comes from cross-group comparisons (See Figure 3).

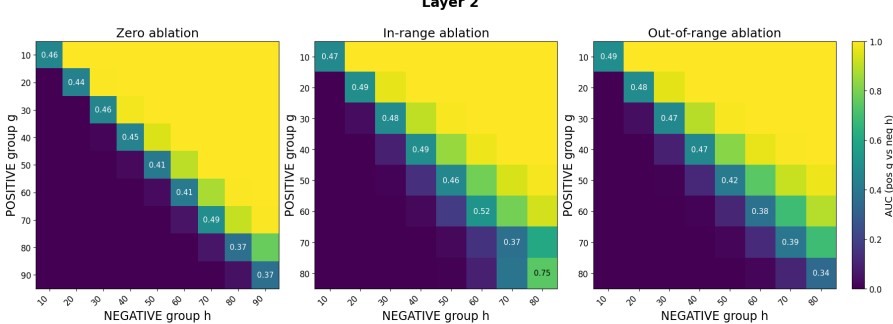

Figure 3: Heatmaps of pairwise AUC for the linearity score ($\alpha = 1$) at Layer 2. Rows (g) are accurate examples at ablation g, columns (h) are inaccurate examples at ablation h; each cell is AUC (positive g vs. negative h). Diagonal numbers show within-ablation separability ( $0.35$–$0.52 \rightarrow$ *weak*), while off-diagonals near 1 indicate the score mainly separates groups at different ablation levels. Panels: zero / in-range / out-of-range corruptions.

Next we evaluated, residual $r(\alpha) = \left| \Delta_{\text{true}}(\alpha) - \Delta_{\text{pred}}(\alpha) \right|$. We observed that $r(\alpha)$ increases monotonically with $\alpha$, and the rise is much steeper on *incorrect* examples than on *correct* ones. By $\alpha=1$ the gap is large, which explains why the $\alpha=1$ residual is a strong per–example predictor. Next, it was seen that for a fixed $\alpha$, $r(\alpha)$ grows smoothly and approximately convexly with ablation percentage; the growth rate itself increases with $\alpha$. A second–order Taylor expansion, implies $r(\alpha) \approx \frac{1}{2}\alpha^2 \left| e^\top H e \right|$, so the observed trends indicate (i) small steps/weak ablations lie in a near–linear regime, while (ii) larger steps/stronger ablations expose curvature along the error direc-

tion ($|e^\top He|$ increases), which is precisely where failures concentrate, linking increased nonlinearity to reduced repairability (See Figure 4). This result holds for both the pretrained models on all layers (See Figure 12) and the NLP task evaluated on an LLM (See Section 4.4). However, unlike the LLMs and other pretrained baselines, our custom 4-layer attention-only model breaks this pattern at the final layer (See Figure 11): that block behaves as a near–hard readout. By the penultimate layer, the correct token is already linearly separable, and the last block primarily rescales/saturates the margin (via LayerNorm + unembedding) rather than "reasoning." In this saturated regime, the clean-point gradient is small or largely orthogonal to the corruption direction, so the scale-test residual no longer tracks repairability and AUC–ROC collapses—even though the model has effectively committed to a prediction one layer earlier.

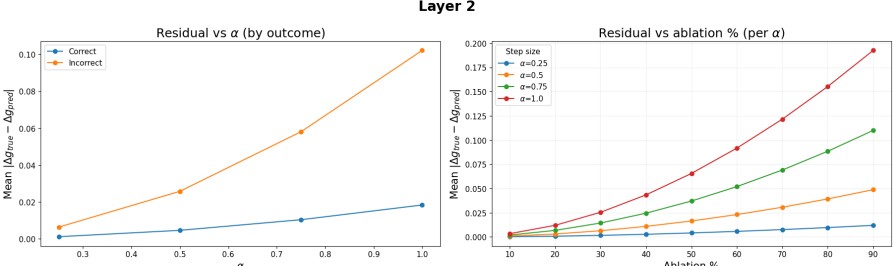

Figure 4: Residual scaling at Layer 2. **Left:** Mean residual vs. step size $\alpha$, split by outcome. Incorrect cases grow much faster, indicating stronger local curvature. **Right:** Residual vs. ablation rate per $\alpha$. Curves rise smoothly (nearly convex) with ablation and steepen with larger $\alpha$, showing that heavier corruption and bigger steps expose more nonlinearity and hence lower repairability.

Per-layer evaluation places the strongest prediction at mid→late depth, coinciding with the *elbow* we observe in repair-difference and token-agreement plots: mid layers gently *damp* the error, while late layers either *amplify* it or enact a sharp correction.

### 4.4 LOCAL LINEARITY BREAKS IN CORRUPTED NLP TASKS

We evaluate three next-token tasks in a unified *prefix → 1-token* format: (i) Subject-Verb agreement (SVA) from BLiMP (Warstadt et al., 2020) by cutting right before `is/are/has/have/does/do`; (ii) Factual cloze from SQuAD (Rajpurkar et al., 2016) with single-word answers (iii) Multiple-choice science QA from the ARC-Challenge dataset (Clark et al., 2018b), where we concatenate the question and its answer options and require the model to predict the single-letter label (A/B/C/D) of the correct choice as the next token. (Clark et al., 2018a). Prefix-only corruptions (word dropout, constant "the") at $p \in \{0, 0.2, 0.4\}$ preserve the label and end with a trailing space. See Appendix A.2.

We run our scaling experiments on three causal LMs- EleutherAI/gpt-neo-1.3B (24 layers) (Gao et al., 2020; Black et al., 2021), openai-community/gpt2-xl (48 layers) (Radford et al., 2019), and microsoft/phi-2 (32 layers) (Javaheripi et al., 2023) evaluating on our SVA, factual cloze, and ARC multiple-choice distractor sets. All three tasks are tested using corruption strengths $p \in 0, 0.2, 0.4$ and attention-drop factors $\alpha \in 0.25, 0.5, 0.75, 1.0$ for every dataset.

Across tasks, predicted vs. patched logit changes are tightly aligned at small $\alpha$ (*Pearson $R \approx$* 0.86–0.88 over all $\alpha$ points). Residuals grow roughly quadratically with $\alpha$ and with ablation strength, indicating curvature along the corruption direction. Depth-wise, early layers are near-linear (tiny residuals), while residuals rise sharply in late layers consistent with mid-depth damping followed by late re-amplification (See Figure 5 and Figure 13, 14 and 15.). Figures show the effect for the majority of evaluated layers under all corruption modes.

### 4.5 CAUSAL INTERVENTION SUGGESTS LINEARITY ENABLES REPAIR

To assess whether local linearity is causally involved in spontaneous context restoration, we perform a controlled manipulation of the LayerNorm scaling parameter $\gamma$. For a hidden vector $h$ with mean $\mu$ and variance $\sigma$, LayerNorm computes

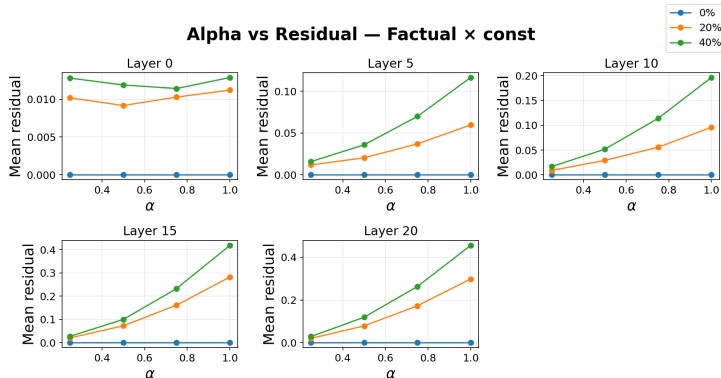

Figure 5: Alpha–residual scaling on Factual (constant corruption) in GPT-Neo model. Mean residual vs. $\alpha$ across layers $\{0, 5, 10, 15, 20\}$ for corruption rates $p \in \{0, 0.2, 0.4\}$. Early layers are near-linear (flat curves), while residuals grow steeply with both $\alpha$ and $p$ in late layers, showing curvature that emerges with depth and corruption.

$$\text{LN}(h) = \gamma \frac{h - \mu}{\sigma} + \beta. \tag{1}$$

The parameter $\gamma$ directly modulates the effective curvature of the representation: decreasing $\gamma$ dampens nonlinear distortions, whereas increasing $\gamma$ amplifies them. Crucially, adjusting $\gamma$ allows us to vary local nonlinearity without modifying any other model weights.

We sweep $\gamma \in \{0.1,\ 0.5,\ 1.5,\ 2.0\}$ for individual decoder layers, keeping all projections and attention weights fixed. This intervention perturbs only the normalization scale of one layer at a time, providing a targeted manipulation of the model's local linearity profile. We recompute Token Agreement (TA) and residual agreement across layers under different corruption strengths and compare these curves to the unmodified model.

Figure 16 shows the effects of varying $\gamma$ in Layers 1 to 3. Reducing the scale ($\gamma = 0.1$ or $0.5$) largely preserves the characteristic mid-layer "repair elbow": agreement initially drops due to error propagation and then recovers in layers 2–3. In contrast, increasing the scale ($\gamma = 1.5$ or $2.0$) significantly disrupts this pattern. Under higher nonlinearity, the agreement peak at the repair layer collapses, and both residual-path and token-level agreements remain suppressed throughout the stack. Because no weights are retrained and only the normalization scale is altered, these effects isolate $\gamma$ and by extension, local linearity as the causal mechanism governing whether a layer can integrate surviving cues. The results therefore demonstrate that linearity is not merely preserved by successful repair; rather, *linearity enables repair*. When $\gamma$ increases and the local trajectory becomes more curved, the model becomes unable to reconstruct the intended representation even though all attention computations remain intact.

### 4.6 TARGETED FINE-TUNING BOOSTS ACCURACY ON CORRUPTED CONTEXT

Starting from our custom base model, we fine–tuned separate copies on a single corruption level $r \in \{10, 30, 50, 90\}\%$ ("FT@$r$"), using only 20% of the training set until convergence, and kept the standard next–token objective (targets are the clean next tokens). We then evaluated each fine–tuned model across the full ablation sweep on held–out data for all three sequence types. The unfine–tuned model is denoted *nil* (See Figure 6). For the fine-tuning purpose, we only used the ZERO-ABLATION regime. Accuracy–vs–ablation curves show consistent gains for FT@10 and FT@30 across tasks, with FT@30 the most robust overall (Fig. 20). In contrast, FT@50 yields mixed effects and FT@90 degrades sharply, indicating over-specialization to heavy corruption and loss of clean/medium–ablation performance. The ($\tau_{50}$, ablation level at 50% accuracy) peaks at FT@30 for CONSTANT and SUBTRACT, with sizable but smaller gains for VAR_ARITHMETIC; beyond FT@30, $\tau_{50}$ collapses (Fig. 20). Mean accuracy shows similar trends (See Figure 19). Thus, light, targeted fine–tuning shifts the entire accuracy curve upward, whereas heavy fine–tuning harms

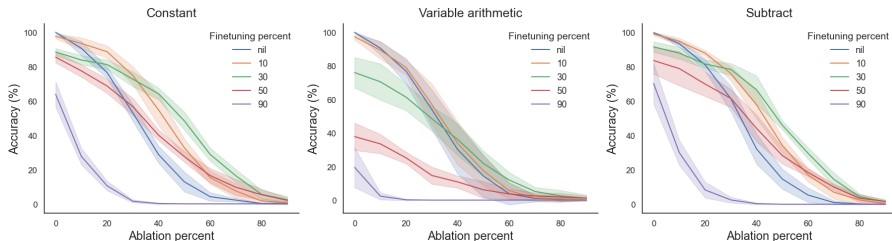

Figure 6: Model accuracy before and after finetuning as a function of ablation level of testing sequence. Each panel shows a different sequence type (left to right: *Constant*, *Variable arithmetic*, *Subtract*). Curves correspond to the fraction of finetuning data used (*nil*, 10%, 30%, 50%, 90%). Lines indicate mean accuracy across runs.

generalization. However, with corrupted ablation, the accuracy on clean tasks decreases from $0.8\%$ (FT@10) to $24\%$ (FT@30).

### 4.7 REPAIR DEPENDS ON RESIDUAL REPRESENTATIONS RATHER THAN SPECIFIC HEADS.

To probe which components of the model contribute to the clean-ablated discrepancy, we perform two types of ablations at the intervention layer (layer 3): head pruning and residual pruning. In all cases, only the *ablated* model is modified; the clean model remains unchanged. We prune different subsets of attention heads at layer 3, ranging from single-head removals to pruning four, six, or all heads. Across all sequence types, the resulting repair–difference curves remain nearly identical to the unpruned case (Figs. 17). Neither the shape nor the magnitude of the repair signal is sensitive to which heads are removed. This indicates that the effect of our corruption is not mediated by a small set of specialized heads at the intervention layer.

We next remove the residual stream at individual layers (Fig. 18). Unlike head pruning, residual pruning produces clear shifts in the repair difference: early-layer pruning (layers 1-2) results in the strongest deviations, while pruning closer to the intervention layer yields a milder but systematic effect. Nevertheless, the characteristic sharp transition between layers 3 and 4 appears in all conditions. These results suggest that the repair phenomenon is not localized to a few heads at the intervention layer but depends more strongly on early residual representations, while the structural transition from layer 3 to layer 4 is remarkably robust to both manipulations.

## 5 DISCUSSION

We find a consistent mid-to-late "elbow" where transformers enact *spontaneous context correction*; this late integration explains *selective robustness* (e.g., SUBTRACT is sturdier under zero masks but falters with in-range decoys) and provides direct evidence of *variable binding*: across heterogeneous corruptions (zero, in-range, out-of-range), models re-instantiate the mapping between roles (slots) and values (numbers) to converge on the same next token Smolensky et al. (2024). However, this is not absolute, as exact programmatic predictors are possible even under heavy corruption. A simple linearity–scale test—comparing first-order logit predictions to patched outcomes predicts repairability: behavior is near-linear (and correctable) on clean or mildly corrupted inputs but becomes curved (and failure-prone) under stronger corruption. These effects replicate across symbolic arithmetic and corrupted NLP, and from a 4-layer transformer to a 2.7B-parameter model.

Two takeaways follow: (i) diagnose at the elbow- use RD/TA to localize the repair step and the linearity residual to forecast success, then fine-tune at *moderate* corruption to lift the whole robustness curve without sacrificing clean performance, while heavy-corruption training over-specializes; (ii) expect portability—the mid-damping/late-correction pattern, local linearity criterion, and variable-binding signature appear architecture- and task-agnostic, so the toolkit should extrapolate to a broad array of LLMs, with the caveat that saturated final layers can obscure linearity signals, so measurements should target the last *decision-forming* block.

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

# A   APPENDIX

## A.1   LLM USAGE

We used a general-purpose large language model only for light writing support (clarity edits, phrasing, and tightening of title/abstract) and minor plotting assistance (suggestions for Matplotlib/LaTeX labels and figure layout). All scientific content, mathematical definitions, code, and figures were verified and version-controlled by the authors, and any LLM-suggested text or code was reviewed and edited for accuracy and reproducibility.

## A.2   TASK CREATION AND CORRUPTIONS FOR NLP OBJECTIVE

**Datasets.** We construct two next-token tasks in a unified *prefix → 1-token target* schema. (i) **SVA (Subject-Verb Agreement):** Across all BLiMP configurations, we cut the prefix immediately before the first auxiliary in {is, are, has, have, does, do}. The target is that auxiliary token. (ii) **Factual cloze:** From SQuAD v1.1 we retain QA pairs with single-word answers and form the prefix "Question:{q} Answer:    "; the target is the answer string. (iii) **ARC multiple-choice QA:** We take the ARC-Challenge split and convert each example into a prefix–target form. Let $q$ be the question and $(\ell_i, c_i)$ the labeled answer choices. We form the prefix: Question: $q$ Choices:  (A) $c_A$ (B) $c_B$ (C) $c_C$ (D) $c_D$ Answer:    (with trailing space). The target is the single-character label of the correct option. For corrupted variants, we prepend $k \in \{0, 1, 2\}$ distractor sentences sampled from a fixed set of factual/common-sense statements according to the ablation level $p \in \{0.0, 0.2, 0.4\}$ (mapping $p=0.2 \to 1$, $p=0.4 \to 2$). Distractors modify only the question text and never the answer choices or the gold label.

**Tokenizer constraints.** We require each target to be a *single tokenizer token* under the evaluation tokenizer (e.g., GPT-2 family). All prompt strings are enforced to end with a trailing space to guarantee the target is the immediate next token.

**Corruption (ablation) operators** Corruptions operate only on the prefix and do not modify the target. We protect the last two prefix tokens to avoid breaking immediate local cues. We apply two simple, label-preserving operators: (a) *word dropout* (delete editable tokens independently with probability $p$), and (b) *constant replacement* (replace editable alphanumeric tokens with the constant "the" with probability $p$). We sweep $p \in \{0.0, 0.2, 0.4\}$ with one random draw per $(p, \text{mode})$ cell.

**Sanity filters.** We drop items that fail any of: (i) non-1-token target, (ii) prefix not ending in a space, (iii) leakage, or (iv) very short prefixes (fewer than three tokens pre-auxiliary for SVA or overall for Factual).

**Sizes and balance.** After corruption, we use approximately $10^3$ examples per task, balanced across ablation levels and corruption modes.

The single-token target lets us define the target logit $g$ precisely at the final prefix position. Prefix-only corruption varies the hidden-state error $e$ while keeping the gold label fixed, enabling our scale test: predicted $\Delta g_{\text{pred}}(\alpha) = \alpha \nabla g^\top e$ vs. patched $\Delta g_{\text{true}}(\alpha)$ over $\alpha \in \{0.25, 0.5, 0.75, 1.0\}$.

**Examples.**

**SVA 1 (plural)**
```
Clean:  The keys to the cabinet
Target:  are
Dropout (p=0.2):  The keys the cabinet
Const (p=0.4):  the keys the the
```
**SVA 2 (singular)**
```
Clean:  The book on the table
Target:  is
Dropout (p=0.2):  The book the table
Const (p=0.4):  the book the the
```
**Factual 1**
```
Clean:  Question:  The capital of France is   Answer:
Target:  Paris
Dropout (p=0.2):  Question:  capital of France   Answer:
Const (p=0.4):  Question:  the the of France the Answer:
```
**Factual 2**
```
Clean:  Question:  The capital of Italy is   Answer:
Target:  Rome
Dropout (p=0.2):  Question:  capital of Italy   Answer:
Const (p=0.4):  Question:  the the of Italy the Answer:
```

## A.3 EXTENDED FIGURES

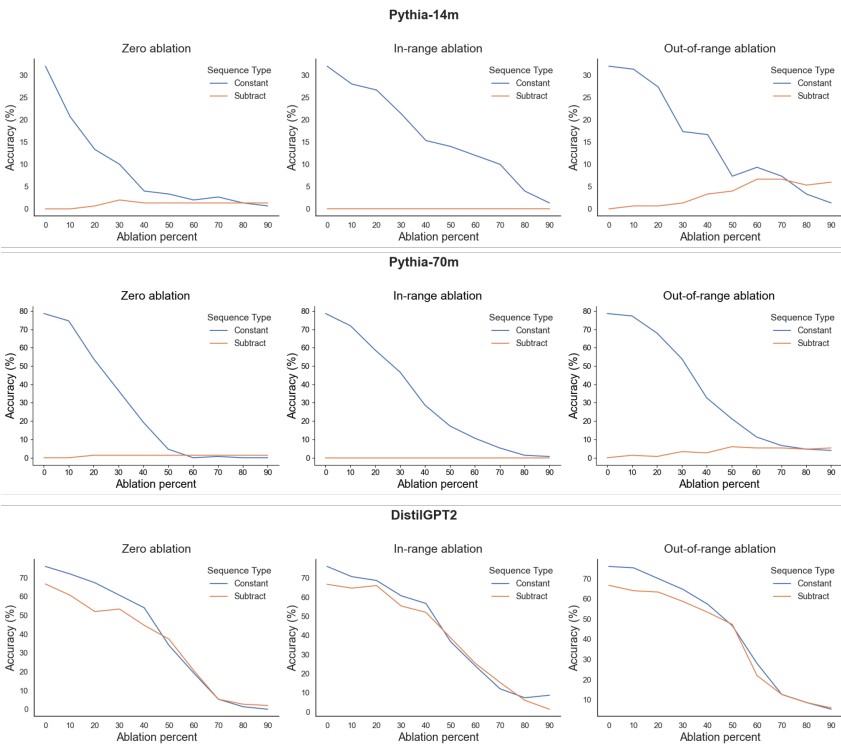

Figure 7: Accuracy of the pre-trained models on three ablation regimes for all three sequence types.

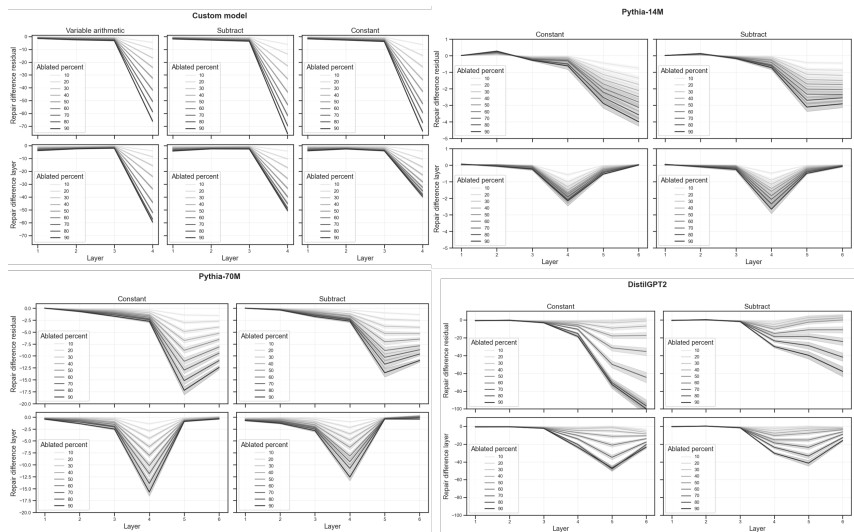

Figure 8: Repair Difference across ablation percentages for all sequence types, across all ablation regimes for all the evaluated models. All figures show *elbow* effect.

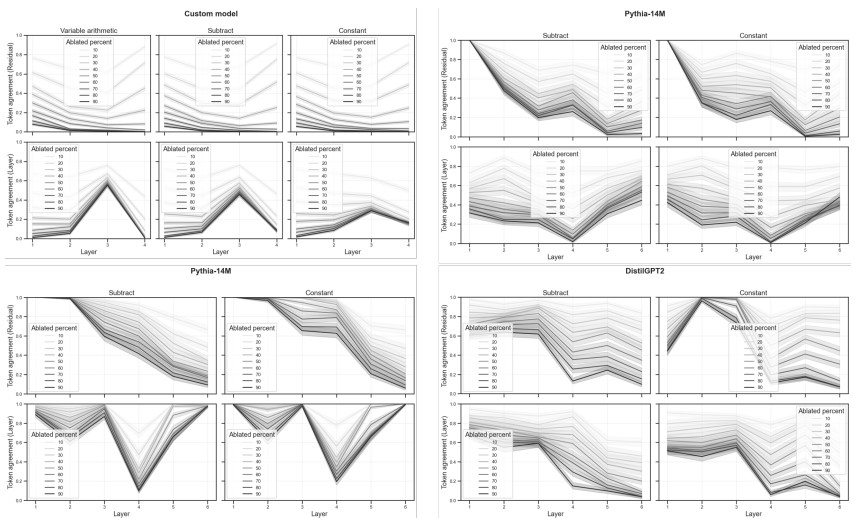

Figure 9: Layer-wise Token Agreement across ablation percentages for all sequence types, across all ablation regimes for all the evaluated models.

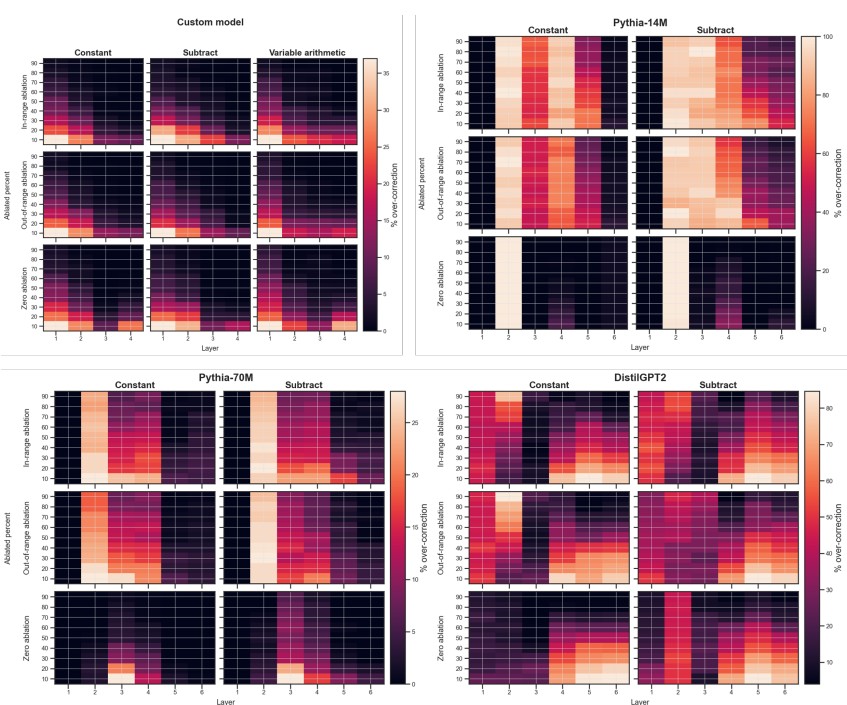

Figure 10: Over-correction across depth and corruption.Heatmaps show the share of cases (color; %) where the ablated run *over-corrects* (RD> 0)—i.e., the correct-token logit on the corrupted prompt exceeds both the clean-run correct-token logit and the best competing token—as a function of layer (x-axis) and ablated percent (y-axis). Columns give sequence families (Constant, Subtract, Variable arithmetic) and panels split models (Custom, Pythia-14M, Pythia-70M, DistilGPT2); within each model the three blocks correspond to zero, in-range, and out-of-range ablations. Over-correction is rare and concentrated in early layers, decaying with depth and with stronger ablation for the custom model, while pretrained models exhibit larger, sometimes late-layer pockets (most prominently in Pythia-14M). Color bar: % over-correction.

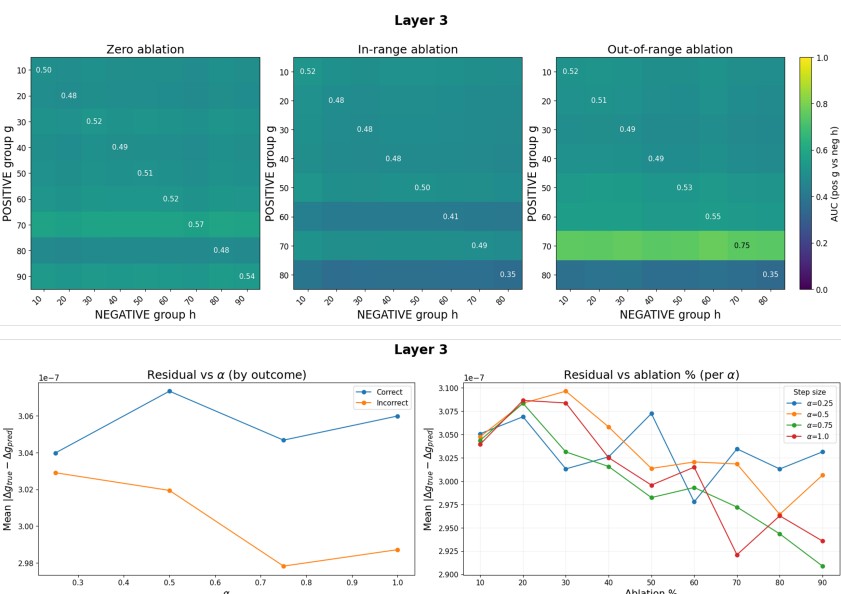

Figure 11: **Top**- Heatmaps of pairwise AUC for the linearity score ($\alpha = 1$) at last layer. Rows (g) are accurate examples at ablation g, columns (h) are inaccurate examples at ablation h; each cell is AUC (positive g vs. negative h). Panels: zero / in-range / out-of-range corruptions. **Bottom**-Residual scaling at the last layer. **Left:** Mean residual vs. step size $\alpha$, split by outcome. **Right:** Residual vs. ablation rate per $\alpha$. Curves rise smoothly (nearly convex) with ablation and steepen with larger $\alpha$, showing that heavier corruption and bigger steps expose more nonlinearity and hence lower repairability.

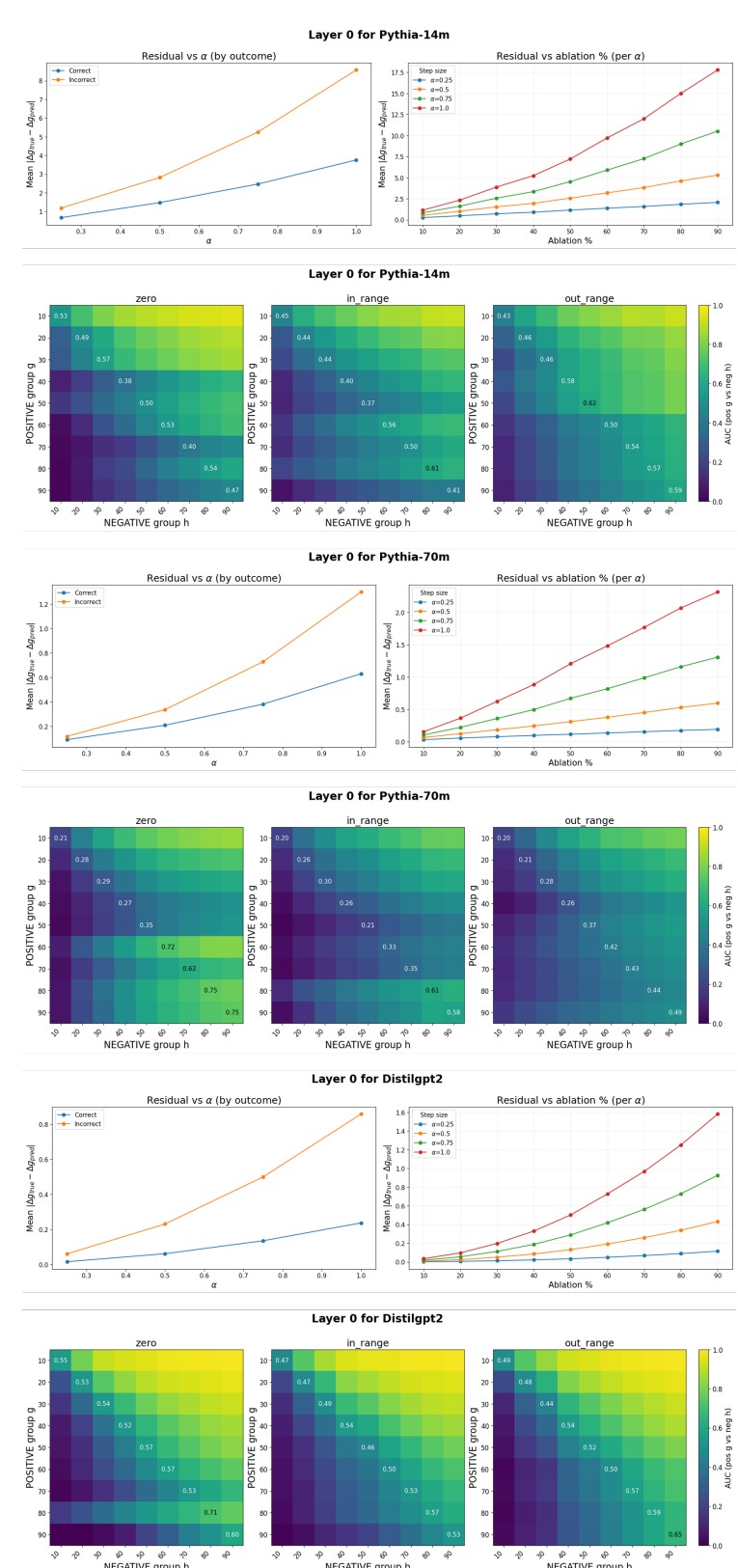

Figure 12: Exemplary heatmaps (at Layer = 0) (see Figure 3) and residual-$\alpha$ curves (See Figure 4) for all the three pre-trained models.

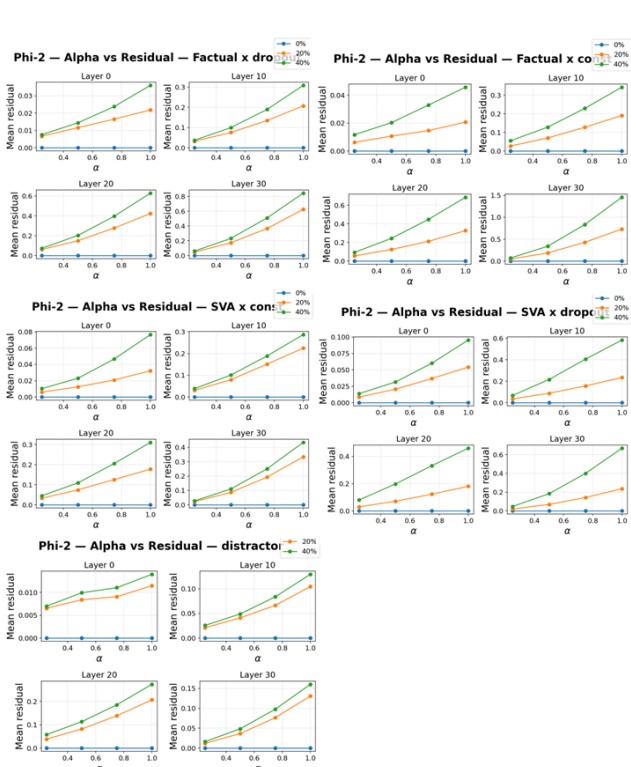

Figure 13: Residual-$\alpha$ curves (See Fiure 5) for different prompt ablation schemes used for the NLP task at different layers for Phi-2

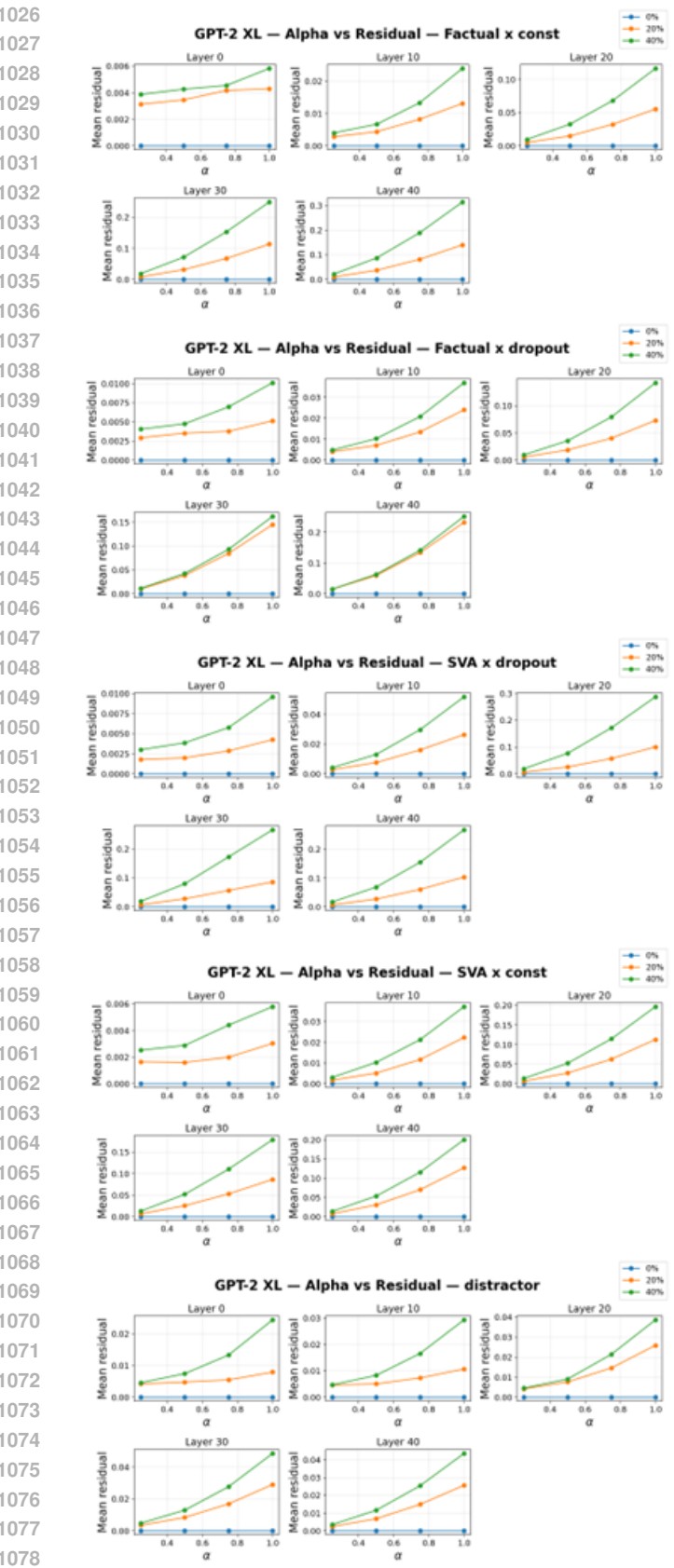

Figure 14: Residual-$\alpha$ curves (See Fiure 5) for different prompt ablation schemes used for the NLP task at different layers for GPT-XL

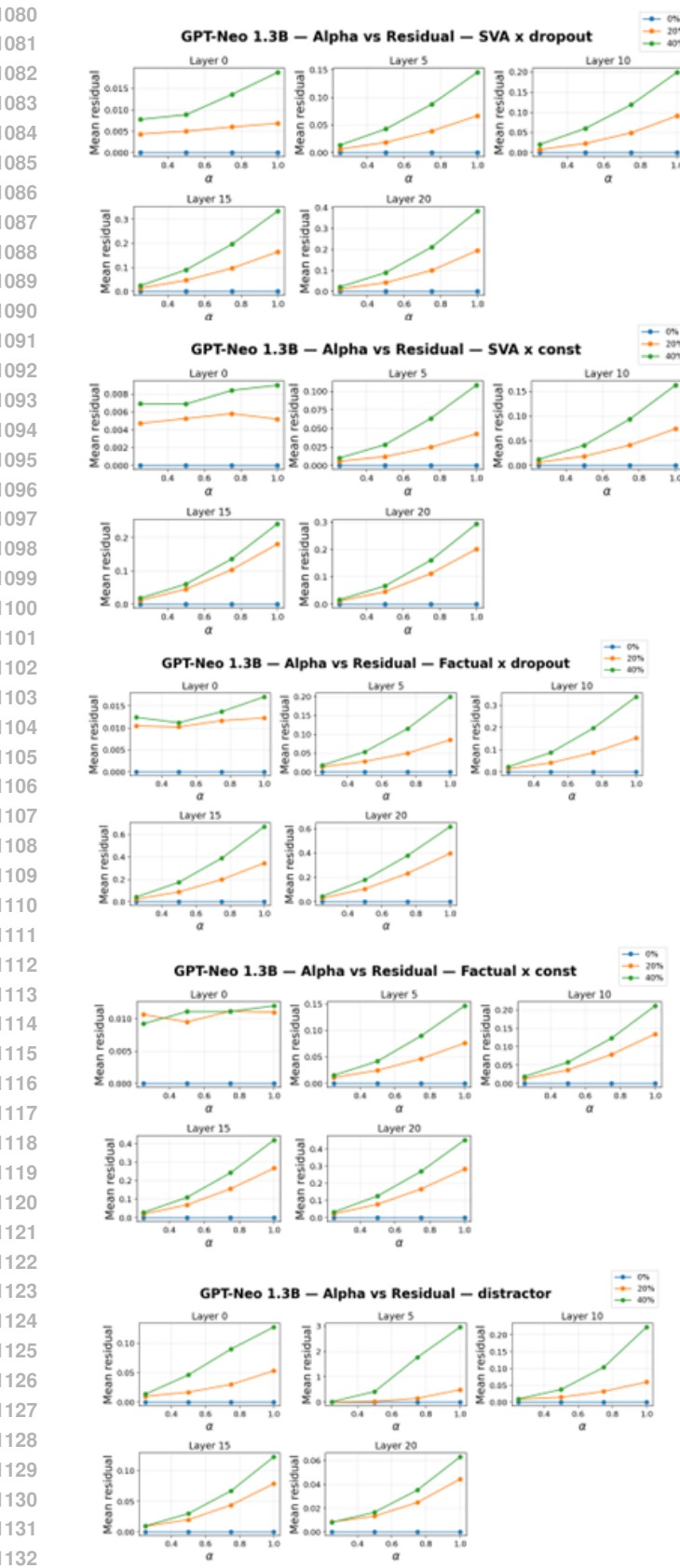

Figure 15: Residual-$\alpha$ curves (See Fiure 5) for different prompt ablation schemes used for the NLP task at different layers in GPT-Neo

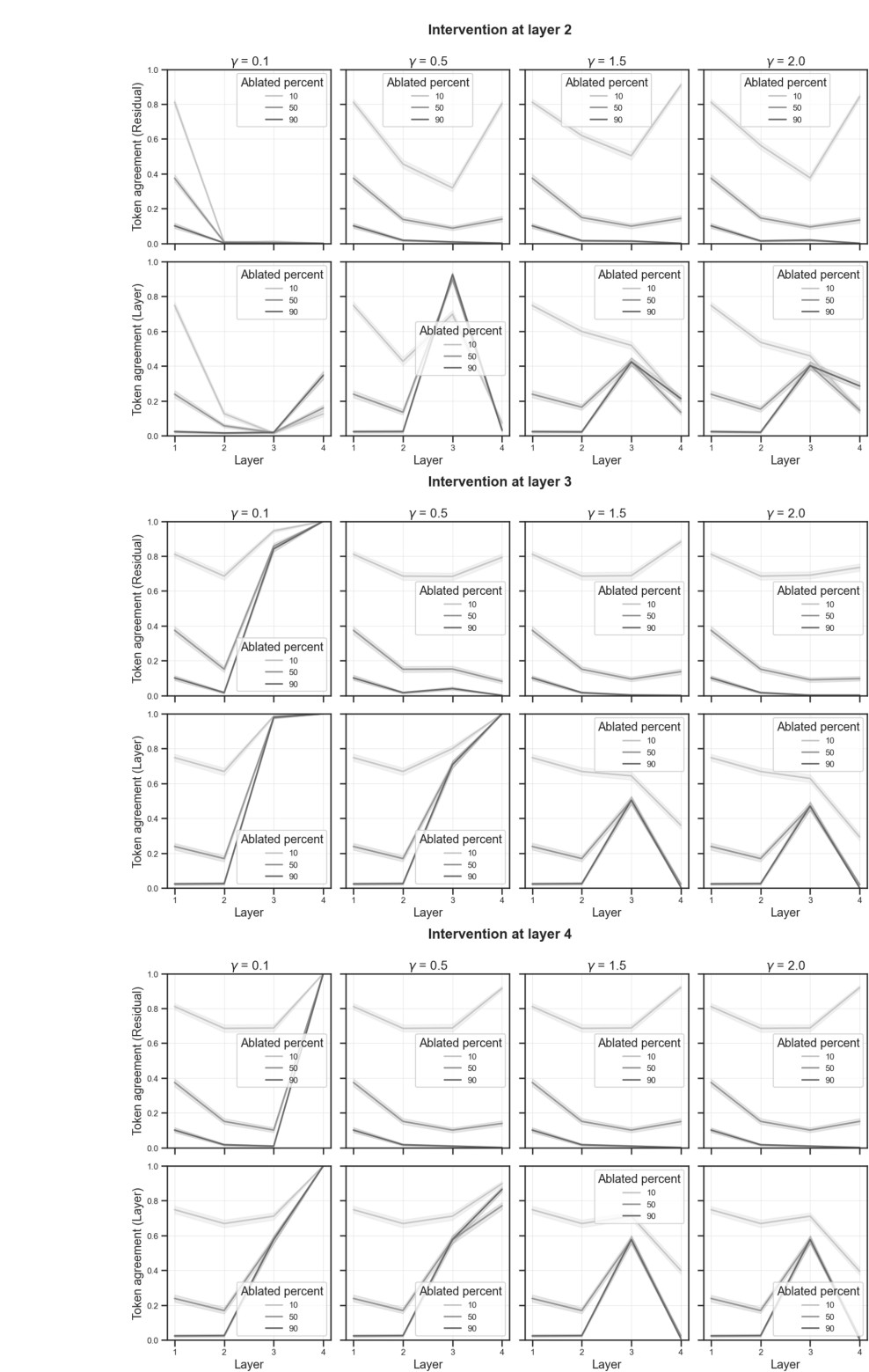

Figure 16: Token Agreement under LayerNorm- $\gamma$ interventions. Lower $\gamma$ (0.1–0.5) preserves the mid-layer repair elbow, while higher $\gamma$ (1.5–2.0) suppresses repair and flattens agreement curves. Since only $\gamma$ is modified, these effects show that increasing local nonlinearity disrupts spontaneous context restoration.

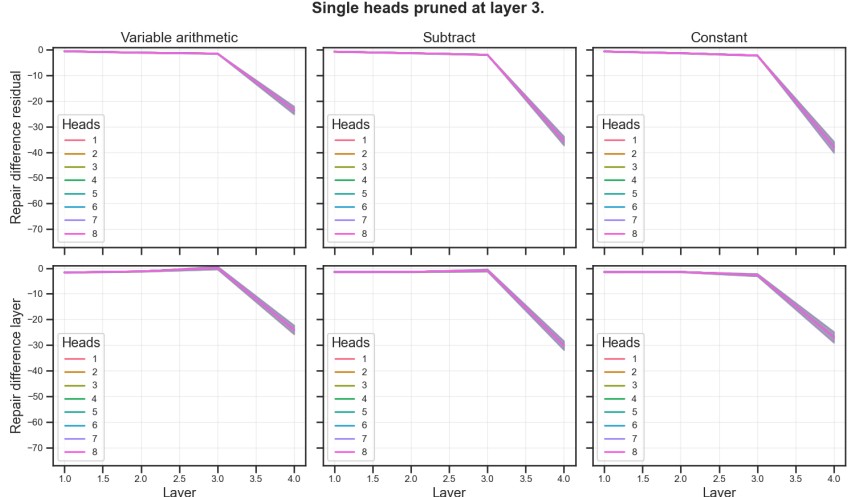

Figure 17: Single-head pruning at layer 3. Pruning individual attention heads has virtually no effect on the clean–ablated repair difference: all curves overlap closely across heads and sequence types. Thus, no single head at the intervention layer is critical for the repair effect, demonstrating strong redundancy across heads.

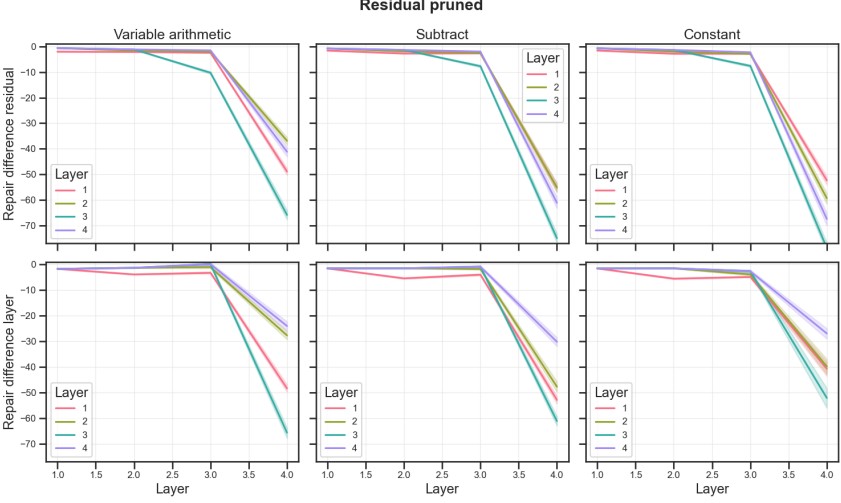

Figure 18: Residual pruning at layers 1–4. Pruning the residual stream produces clear shifts in the clean–ablated repair difference: earlier-layer pruning leads to larger deviations, while pruning closer to the intervention layer yields a milder but systematic effect. Across all conditions, the sharp transition between layers 3 and 4 persists, indicating that the core repair phenomenon is structurally robust even when the residual pathway is ablated.

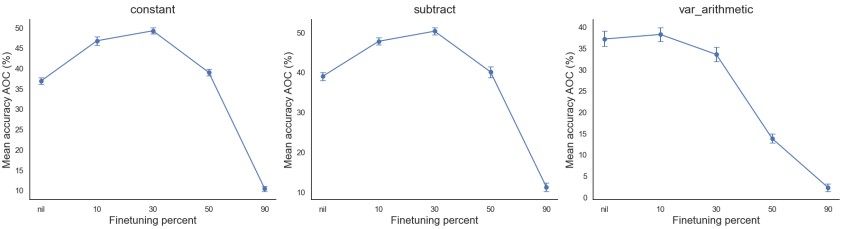

Figure 19: Mean accuracy AOC (%): average accuracy across ablation levels as a function of finetuning percent for three sequence types (*constant*, *subtract*, *variable-arithmetic*). Larger AOC indicates better overall performance under ablation.

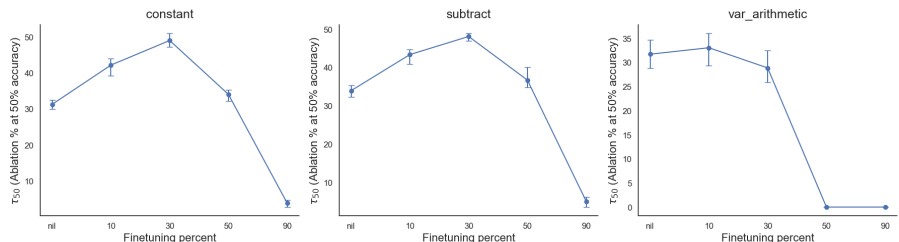

Figure 20: Estimated $\tau_{50}$ as a function of finetuning percent for three sequence types (*constant*, *subtract*, *variable-arithmetic*). Larger $\tau_{50}$ indicates greater robustness to ablation.

