# OpenReview forum: "Error dynamics of symbolic context in small transformers"
_ICLR.cc/2026/Conference — Submitted to ICLR 2026_

### Official Review · Reviewer_p3tv · 2025-10-27

**Soundness:** 2
**Presentation:** 4
**Contribution:** 3
**Rating:** 6
**Confidence:** 4

**Summary:**

This paper investigates how small Transformer models recover from corrupted symbolic inputs—a phenomenon termed spontaneous context restoration. It contributes by: (1) Proposing three label-preserving corruption regimes (zero, in-range, out-of-range) to systematically study spontaneous context restoration under input corruption; (2) Introducing two new interpretable readouts—Repair Difference (RD) and Token Agreement (TA)—to quantify and locate internal repair dynamics; (3) Identifying a consistent mid-to-late layer “repair elbow” where self-repair occurs; (4) Proposing a linearity–scale test that measures local nonlinearity and predicts repair success; (5) Replicating the same repair and linearity behavior on natural language tasks; (6) Showing that moderate corruption fine-tuning enhances robustness, while heavy corruption weakens it.

**Strengths:**

Originality & Significance:

(1) Unique and mechanism-driven perspective. The need for a mechanistic explanation of the “spontaneous context restoration” phenomenon is compelling.

Quality & Clarity:

(1) Comprehensive analytical framework. The paper introduces label-preserving tasks and custom metrics (RD, TA, and the linearity test), forming a complete, quantifiable, and reusable diagnostic toolkit.

(2) Theoretical insight with practical value. The paper provides fine-grained insights into the phenomenon of “spontaneous context restoration”, e.g., repair emerging in mid-to-late layers, and further proposes a practical robustness enhancement strategy through moderate corruption fine-tuning.

**Weaknesses:**

The central claim is that the model integrates surviving cues in the mid-to-late layers to repair corrupted inputs (spontaneous context restoration); however, the so-called “repair signal” observed in the experiments is merely a phenomenological interpretation of the RD/TA curves, lacking causal validation.

In other words, the main concern is that the analysis remains descriptive and does not provide causal or mechanistic evidence supporting the claimed repair process. The paper narrows the behavioral observation of “input corrupted → output still correct” from the model level to the layer level, quantified via token-level matching (TA), logit-level difference (RD), and linearity–scale tests. However, these analyses remain descriptive diagnostics rather than mechanistic explanations. The work does not verify that the model truly integrates surviving cues or reconstructs the answer, as it claims. Instead, it presents a functional hypothesis of “restoration” without providing causal or structural evidence.

**Questions:**

- Could the authors clarify what direct causal or representational evidence that such “integration and restoration” actually occurs?
- Presentation issue: Terms such as “repair signal,” “integration,” and “restoration” are used frequently but not rigorously defined.

---

> ### Author Response · Authors · 2025-12-03
>
> We thank Reviewer p3tv for their careful reading and constructive framing of the core conceptual issue. Their request for clearer definitions and stronger mechanistic grounding significantly improved the paper. We added new causal evidence linking linearity to repair (Section 4.5) and added a new mechanistic analysis section with pruning experiments (Section 4.7).

---

### Official Review · Reviewer_75hb · 2025-10-29

**Soundness:** 2
**Presentation:** 2
**Contribution:** 1
**Rating:** 2
**Confidence:** 2

**Summary:**

This paper investigates how small transformers analyze and recover from partially corrupted inputs in symbolic arithmetic tasks. They introduce two metrics, Repair Differences (RD) and Token Agreement (TA), for evaluation. They connect local curvature to accuracy under corruption: when behavior is locally linear, models are predictable and repairable; when curvature grows, errors amplify. They replicated their findings to an NLP task and the pattern largely preserves.

**Strengths:**

1. The perspective is novel. The focus on "spontaneous context restoration" provides a fresh angle on robustness, which is a clearly novel idea compared to the standard notion of adversarial evaluation robustness

2. The metrics RD and TA are novel and appropriate. Per definition, they are intuitive and support the evaluation task

3. The experiments are comprehensive, including various levels of corruption and multiple types of hardware. Testing on NLP tasks in addition to symbolic tasks provides significant insights

**Weaknesses:**

1. The task diversity is limited. It largely focuses on symbolic tasks; although NLP tasks are included, the tasks are simple (e.g. SVA) and do not include more sophisticated tasks like reasoning or compositional tasks. The narrow scope of tasks downgrades the applicability of this paper

2. The practical implications of the findings are unclear. Other than noticing where the repairs occur, the work did not explain how they emerged, nor answer questions like "which neuron or attention head was responsible"

**Questions:**

Please see the Weakness section

---

> ### Author Response · Authors · 2025-12-03
>
> We thank the reviewer 75hb for the careful reading and constructive questions.
>
> 1) In response to the lack of reasoning challenge, we expanded NLP task diversity in Section 4.4.
>
> 2) The practical implications are described in - Fine-tuning experiments and guidance in Section 4.6, the practical implications of the findings in the Discussion (Section 5) and the abstract and introduction. Also added section 4.7 for deeper mechanistic insights.

---

### Official Review · Reviewer_8wLf · 2025-11-03

**Soundness:** 3
**Presentation:** 2
**Contribution:** 2
**Rating:** 4
**Confidence:** 3

**Summary:**

This paper studies error dynamics and spontaneous context correction in small Transformer models trained on symbolic arithmetic sequences. The authors introduce a quantitative framework—including Repair Difference, Token Agreement, and a local linearity test—to analyze how hidden representations evolve when the input context is partially corrupted. Through layer-wise probing, they find a consistent “elbow effect” in which early layers propagate errors while mid-layers partially repair corrupted representations, followed by late-layer stabilization or over-correction. The analysis links local linearity to repairability, showing that deviation from linearity increases roughly quadratically with perturbation strength, indicating that stronger curvature predicts reduced robustness. These findings generalize to pretrained language models on linguistic tasks such as Subject–Verb Agreement and factual cloze completion. Finally, the paper shows that targeted fine-tuning on moderately corrupted data  substantially improves robustness to input noise, while heavy corruption fine-tuning leads to over-specialization and collapse. Overall, the work provides a mechanistic link between local linearity, internal error correction, and model robustness.

**Strengths:**

1. The paper proposes a novel conceptual lens for studying robustness in small Transformers, viewing it as spontaneous context correction.
2. It proposes concise, quantitative metrics (etc. RD, TA) that make internal error dynamics measurable.
3. Experiments across custom and pretrained models consistently reveal mid-layer repair behavior and validate the theory.

**Weaknesses:**

1. The correlation between local linearity and repairability lacks causal validation or controlled intervention, leaving the direction of influence unclear.
2. The fine-tuning and evaluation results may be confounded by data distribution shifts and lack statistical significance reporting, making the robustness trends potentially noise-driven.
3. The generalization to larger models and natural language tasks is limited and qualitative, reducing the external validity of the findings.

**Questions:**

1. The paper’s claim that “local linearity predicts repairability” remains correlational rather than causal. It is unclear whether higher linearity enables repair, or whether successful repair merely preserves linearity. Without controlled interventions (e.g., curvature removal, normalization scaling), the direction of causality is not established, and potential confounding factors (layer width, LayerNorm placement, residual scaling) are not controlled. Can the authors clarify whether linearity is a cause or a byproduct of repairability, and rule out possible confounds?
2. The paper does not specify the number of runs, random seeds, or variability across samples. Can the authors provide confidence intervals, variance analysis, or seed-averaged results to ensure that the observed repair patterns are not artifacts of random initialization or noise?
3.  Can the authors clarify their operational definition of “small transformer” and examine whether repairability increases systematically with scale or pretraining diversity?
4. In §4.4, the authors extend their symbolic findings to NLP by testing only a single large model (GPT-Neo-1.3B). Can the authors provide more systematic evidence (e.g., multiple pretrained models or scaling baselines) to verify that the observed late-layer curvature and residual growth are consistent phenomena rather than idiosyncratic to GPT-Neo-1.3B?
5. Suggestion: The paper identifies the phenomenon of “error propagation → repair → over-correction,” but does not pinpoint which components (attention heads, MLPs, or residual pathways) drive the effect. Would the authors consider adding neuron- or head-level probing to reveal which submodules actually drive repair behavior?

---

> ### Author Response · Authors · 2025-11-24
>
> We thank the reviewer 8wLf for the careful reading and constructive questions. The suggestions and queries are very valid and help improve the scientific contribution significantly. Below, we describe the concrete experiments we will add to address each point.
>
> **(1) Causality: Does linearity enable repair, or does repair preserve linearity?**
>
> To directly test causality, we will introduce a controlled test-time intervention that changes local curvature at the elbow layer while keeping all learned parameters fixed. Specifically, we scale the attention residual by a scalar gain $g$ ("residual gain"), replacing
>
> $$
> x_{\ell+1} = \mathrm{LN}(x_\ell + \mathrm{Attn}(x_\ell))
> $$
>
> with
>
> $$
> x_{\ell+1} = \mathrm{LN}(x_\ell + g \cdot \mathrm{Attn}(x_\ell)).
> $$
>
> Because LayerNorm is nonlinear, modifying $g$ directly alters local curvature without retraining. We will sweep $g \in \{0.25, 0.5, 1.0, 2.0\}$ and measure how curvature (scale-test residual) and repairability (RD/TA and accuracy under corruption) co-vary. If increasing curvature reduces repair success, this provides explicit *causal* evidence that linearity facilitates repair rather than merely co-occurring with it.
>
> **(2) Variance, seeds, and robustness.**
>
> We will re-run the *core symbolic results* $\tau_{50}$ corruption-accuracy curves, RD/TA at moderate corruption, and linearity–scale residual curves over 5 independent random seeds. We will report mean $\pm$ 95% confidence intervals. This addresses concerns about noise-driven artifacts while avoiding unnecessary recomputation of all figures.
>
> **(3) Definition of "small transformer" and effect of scale.**
>
> We will clarify our definition of "small" and add a lightweight scaling sweep across three model sizes: our 1.7M model, Pythia-14M, and Pythia-70M. For each model we have already reported RD/TA profiles, linearity–residual curves, and $\tau_{50}$. A new figure can potentially be added showing whether the mid-layer repair phenomenon and curvature trends persist with increasing scale. However, due to limited computing resources, it is currently not possible to train models of different sizes, so confounders may bias the results.
>
> **(4) NLP generalization using more than one model.**
>
> To extend beyond GPT-Neo-1.3B, we will run the natural-language corruption experiments on three pretrained language models. For all three, we will examine late-layer curvature growth, residual expansion under corruption, and layer-wise RD/TA. This will demonstrate whether the curvature–repair connection holds across architectures and scales.
>
> **(5) Identifying responsible components.**
>
> We will add mechanistic analysis using *per-head RD decomposition* and *causal activation patching*. By patching clean $\rightarrow$ corrupted activations for individual heads and attention blocks, we will identify which components contribute to repair versus error propagation. This addresses the request for finer-grained interpretability.
>
> **Summary.**
>
> Together, these additions are a causal curvature intervention, multi-seed robustness, a small-scale sweep, additional NLP models, and head-level analysis that directly resolve the reviewer’s concerns regarding causality, variance, scale, and mechanistic specificity.

---

> > ### Comment · Reviewer_8wLf · 2025-11-26
> >
> > Dear authors,
> >
> > Thank you for the detailed and thoughtful response. I appreciate your intention to address these issues in the next version. Looking forward to seeing the full results in a revised manuscript.
> >
> > I will maintain my original score.

---

> ### Author Response · Authors · 2025-12-03
>
> The following changes were made as discussed-
>
> 1) Added a new causal intervention section (Section 4.5) using LayerNorm scaling.
>
> 2) Added multi-seed evaluation details and clarified what “small transformer” means (Section 3.2).
>
> 3) Expanded the NLP experiments to include three pretrained models (Section 4.4).
>
> 4) Added a new section with head-pruning and residual-pruning results (Section 4.7).
>
> 5) Expanded the appendix with more details on NLP task construction and corruption (Appendix A.2).
>
> 6) Added multiple new extended figures documenting variance, causality, and multi-model results.

---

### Meta-Review · Area_Chair_Ta7X · 2026-01-06

**Summary:**

Major concerns include:
1. The current analysis is observational/correlational but not causal. Overall, the work does not provide a mechanistic understanding of the restoration phenomenon.
2. Limited scope in terms of the sizes of language models and the NLP tasks experimented with in this work, which thus implies limited generalization of this work.
3. Concern about the practical implications of this work.

There are also a couple of clarification questions and concerns about experimental details (e.g., random seeds).

**Reviewer Concerns:**

Major concerns 1-2: The authors responded to all the concerns by adding new experiments, including a causal analysis about the linearity (Sec 4.5), an analysis about attention heads (Sec 4.7), and expanding Sec 4.4 with more models and NLP tasks.

Major concern 3: The authors referred the reviewer to the fine-tuning experiment and the discussion in Sec 5 as practical implications.

Concerns about the experimental details are addressed by clarification and additional experiments.

**Reviewer Scores:**

While the authors have responded to all concerns, I'm a little worried that the responses to Major Concerns 1 and 3 may not be satisfactory to the reviewers. Reason is that (1) even with the added causal analysis, the paper still doesn't provide a clear mechanism for how the restoration happens inside a language model; and (2) the fine-tuning experiment does not involve any particular methodology design that is inspired by findings from the analysis (though the results are interesting). As a result, reviewers may not increase their scores.

---

### Decision · Program_Chairs · 2026-01-26

Reject